# Allele imputation for the killer cell immunoglobulin-like receptor KIR3DL1/S1

**Genelle F. Harrison**[1,2☺], **Laura Ann Leaton**[1,2☺], **Erica A. Harrison**[3], **Katherine M. Kichula**[1,2], **Marte K. Viken**[4,5], **Jonathan Shortt**[1], **Christopher R. Gignoux**[1], **Benedicte A. Lie**[4,5], **Damjan Vukcevic**[6,7], **Stephen Leslie**[6,7,8], **Paul J. Norman**[1,2]\*

**1** Division of Biomedical Informatics and Personalized Medicine, University of Colorado, Anschutz Medical Campus, Aurora, Colorado, United States of America, **2** Department of Immunology and Microbiology, University of Colorado, Anschutz Medical Campus, Aurora, Colorado, United States of America, **3** Independent Researcher, Broomfield, Colorado, United States of America, **4** Department of Immunology, University of Oslo and Oslo University Hospital, Oslo, Norway, **5** Department of Medical Genetics, University of Oslo and Oslo University Hospital, Oslo, Norway, **6** School of Mathematics and Statistics, University of Melbourne, Parkville, Victoria, Australia, **7** Melbourne Integrative Genomics, University of Melbourne, Parkville, Victoria, Australia, **8** School of BioSciences, University of Melbourne, Parkville, Victoria, Australia

☺ These authors contributed equally to this work.
\* paul.norman@cuanschutz.edu

**Data Availability Statement:** All code written in support of this publication, imputation models, test data and documentation on installing and running

## Abstract

Highly polymorphic interaction of KIR3DL1 and KIR3DS1 with HLA class I ligands modulates the effector functions of natural killer (NK) cells and some T cells. This genetically determined diversity affects severity of infections, immune-mediated diseases, and some cancers, and impacts the course of immunotherapies, including transplantation. KIR3DL1 is an inhibitory receptor, and KIR3DS1 is an activating receptor encoded by the *KIR3DL1/S1* gene that has more than 200 diverse and divergent alleles. Determination of *KIR3DL1/S1* genotypes for medical application is hampered by complex sequence and structural variation, requiring targeted approaches to generate and analyze high-resolution allele data. To overcome these obstacles, we developed and optimized a model for imputing *KIR3DL1/S1* alleles at high-resolution from whole-genome SNP data. We designed the model to represent a substantial component of human genetic diversity. Our Global imputation model is effective at genotyping *KIR3DL1/S1* alleles with an accuracy ranging from 88% in Africans to 97% in East Asians, with mean specificity of 99% and sensitivity of 95% for alleles >1% frequency. We used the established algorithm of the HIBAG program, in a modification named Pulling Out Natural killer cell Genomics (PONG). Because HIBAG was designed to impute *HLA* alleles also from whole-genome SNP data, PONG allows combinatorial diversity of *KIR3DL1/S1* with *HLA-A* and *-B* to be analyzed using complementary techniques on a single data source. The use of PONG thus negates the need for targeted sequencing data in very large-scale association studies where such methods might not be tractable.

are publicly available at https://github.com/NormanLabUCD/PONG.

**Funding:** This study was performed with support from National Institutes of Health of the USA, R56 AI151549 (PJN) and R01 AI128775 (PJN), and R01 HG010297 (CRG). Australian NH and MRC, 2004262 (SL). The funders had no role in study design, data collection and analysis, decision to publish, or preparation of the manuscript.

**Competing interests:** I have read the journal's policy and the authors of this manuscript have the following competing interests: Dr. Stephen Leslie is a partner with Peptide Groove LLP. All other authors have no competing interests to declare.

## Author summary

Natural killer (NK) cells are cytotoxic lymphocytes that identify and kill infected or malignant cells and guide immune responses. The effector functions of NK cells are modulated through polymorphic interactions of KIR3DL1/S1 on their surface with the human leukocyte antigens (HLA) that are found on most other cell types in the body. KIR3DL1/S1 is highly polymorphic and differentiated across human populations, affecting susceptibility and course of multiple immune-mediated diseases and their treatments. Genotyping *KIR3DL1/S1* for direct medical application or research has been encumbered by the complex sequence and structural variation, which requires targeted approaches and extensive domain expertise to generate and validate high-resolution allele calls. We therefore developed Pulling Out Natural Killer Cell Genomics (PONG) to impute *KIR3DL1/S1* alleles from whole genome SNP data, and which we implemented as an open-source R package. We assessed imputation performance using data from five broad population groups that represent a substantial portion of human genetic diversity. We can impute *KIR3DL1/S1* alleles with an accuracy ranging from 88% in Africans to 97% in East Asians. Globally, imputation of *KIR3DL1/S1* alleles having frequency >1% has a mean sensitivity of 95% and specificity of 99%. Thus, the PONG method both enables highly sensitive individual-level calling and makes large scale medical genetic studies of *KIR3DL1/S1* possible.

## Introduction

The *KIR3DL1/S1* gene encodes highly polymorphic receptors that are expressed by natural killer (NK) cells and some T cells to modulate their effector functions in immunity [1, 2]. The receptors interact with HLA class I ligands that are expressed by most nucleated cells to signify their health status to the immune system [3, 4]. KIR3DL1 allotypes are inhibitory receptors, specific for subsets of highly polymorphic HLA-A and B [5, 6]. The KIR3DS1 allotypes are activating receptors, specific for non-polymorphic HLA-F and a smaller subset of HLA-A and -B [7–9]. Sequence diversity of KIR3DL1/S1 and HLA class I allotypes diversifies human immune responses to specific infections, cancers, cancer treatment and transplantation [10–19]. Accordingly, this genetically determined diversity also associates with differential susceptibility and severity for multiple immune-mediated diseases [20–26]. Although these factors render it imperative to genotype *KIR3DL1/S1* allotypes accurately for medical research and applications that include therapy decisions [27, 28], the high complexity of the genomic region presents obstacles for standard ascertainment methods [29]. The ability to impute alleles from whole-genome SNP genotype (WG-SNP) data will decrease expense and effort, and greatly increase the capacity of research or applications where knowledge of KIR3DL1/S1 and HLA class I combinatorial diversity is critical.

The *KIR* locus, on human chromosome 19, is highly divergent in sequence and structure [29]. As defined by the extensively curated ImmunoPolymorphism Database (IPD), *KIR3DL1/S1* has 220 alleles characterized (release 2.10.0: December 2020), with large numbers continuing to be discovered [30]. As observed for polymorphic *HLA*, the *KIR3DL1/S1* alleles both distinguish individuals and characterize broad ancestral human populations [31, 32]. As a likely consequence of selective pressure providing resistance to infectious diseases [33], specific combinations of KIR3DL1/S1 and HLA associate, differentially across populations, with severity of specific viral infections or autoimmune diseases [34–41]. Likewise, specific combinations of KIR3DL1/S1 with HLA class I influence cancer susceptibilities non-uniformly across populations [42]. In this regard, two key areas of human health significantly impacted by the

population differentiation of KIR3DL1/S1 combinatorial diversity with HLA are HIV research and treatment, and cancer therapy [17, 43–48]. In particular, specific combinations of KIR3DL1/S1 and HLA allotypes influence rejection and relapse rates following transplantation [49–52]. For these reasons, it is critical to establish methods for elucidating genetic variation in *KIR3DL1/S1* that can accommodate the full range of human genetic diversity.

KIR3DL1 specifically binds to subsets of HLA-A or B that carry a five amino acid motif, termed Bw4, on their external facing α1-helix [53]. Expression of KIR3DL1 gives NK cells the ability to detect diseased cells that may have lost or altered expression of these HLA class I molecules [54, 55]. KIR3DL1 also likely serves as an immune checkpoint inhibitor for functionally mature T cells [56]. KIR3DL1 polymorphism, and polymorphism both within and outside the Bw4 motif of HLA affects the specificity and strength of the interaction [57–59]. Polymorphism also determines the expression level or signal transduction abilities of the receptor [60, 61]. KIR3DL1/S1 segregates into three ancient lineages (015, 005 and 3DS1) that have distinct expression and function phenotypes [31]. The 015 lineage comprises inhibitory receptors having high expression and high affinity for Bw4⁺HLA-B. The 005 lineage are inhibitory receptors having low expression and preferential affinity for Bw4⁺HLA-A. The 3DS1-lineage are activating receptors specific for HLA-F and some Bw4⁺HLA-B allotypes expressed by infected cells [62–64]. As defined by these phenotypes, the lineages differentially associate with distinct pathological phenotypes [15, 65–67]. Exceptions to these broad rules (e.g. 3DL1*007 belongs to the 015 lineage but has low expression [61]) contribute to a hierarchy of receptor allotype strengths and reinforce the need to genotype *KIR3DL1/S1* to high resolution [68–70].

Multiple methods are available to impute *HLA class I* genotypes with high accuracy from WG-SNP data [71–75]. We chose to adapt one of these programs so that *KIR3DL1/S1* and *HLA-A* and *B* genotypes could be imputed from the same data source, using an identical algorithm. In the current study we have adapted the HIBAG framework [75] to impute *KIR3DL1/S1* alleles, in a modification we have named <u>P</u>ulling <u>O</u>ut <u>N</u>K cell <u>G</u>enomics (PONG). There are two components to the process: 1) model building that employs machine-learning to determine which combinations of SNPs correlate with known alleles, and 2) imputation that uses this model to determine *KIR3DL1/S1* allele genotypes from study cohorts [75, 76]. Construction of the imputation models required high-resolution *KIR3DL1/S1* alleles and WG-SNP data obtained from the same set of individuals. PONG thus serves as a complement to PING (Pushing Immunogenetics to the Next Generation), which can determine *KIR3DL1/S1* alleles from high throughput sequence data [77]. With a goal to create a model representing a substantial component of human genetic diversity, we compiled and rigorously tested the imputation using data from the 1,000 Genomes populations [78]. The R package PONG is freely available, as are the data sets and imputation models described in this study.

## Materials and methods

### Method overview

*KIR3DL1/S1* exhibits exceptional sequence polymorphism as well as variation in gene content (Fig 1A). Here, we adapted and optimized the framework of HIBAG [75] to impute *KIR3DL1/S1* alleles, in a modification we have named Pulling Out NK cell Genomics (PONG). The development of PONG was focused on building a robust training model that could be used to impute unknown *KIR3DL1/S1* alleles from WG-SNP data across diverse global populations. Training an imputation model requires an input of individuals having known *KIR3DL1/S1* genotypes, coupled with high-density SNP data from the *KIR* region, as typically obtained through whole-genome SNP analysis (Fig 1B). We optimized the process using 1,000 Genomes individuals, because we had previously determined their *KIR3DL1/S1* alleles from sequence

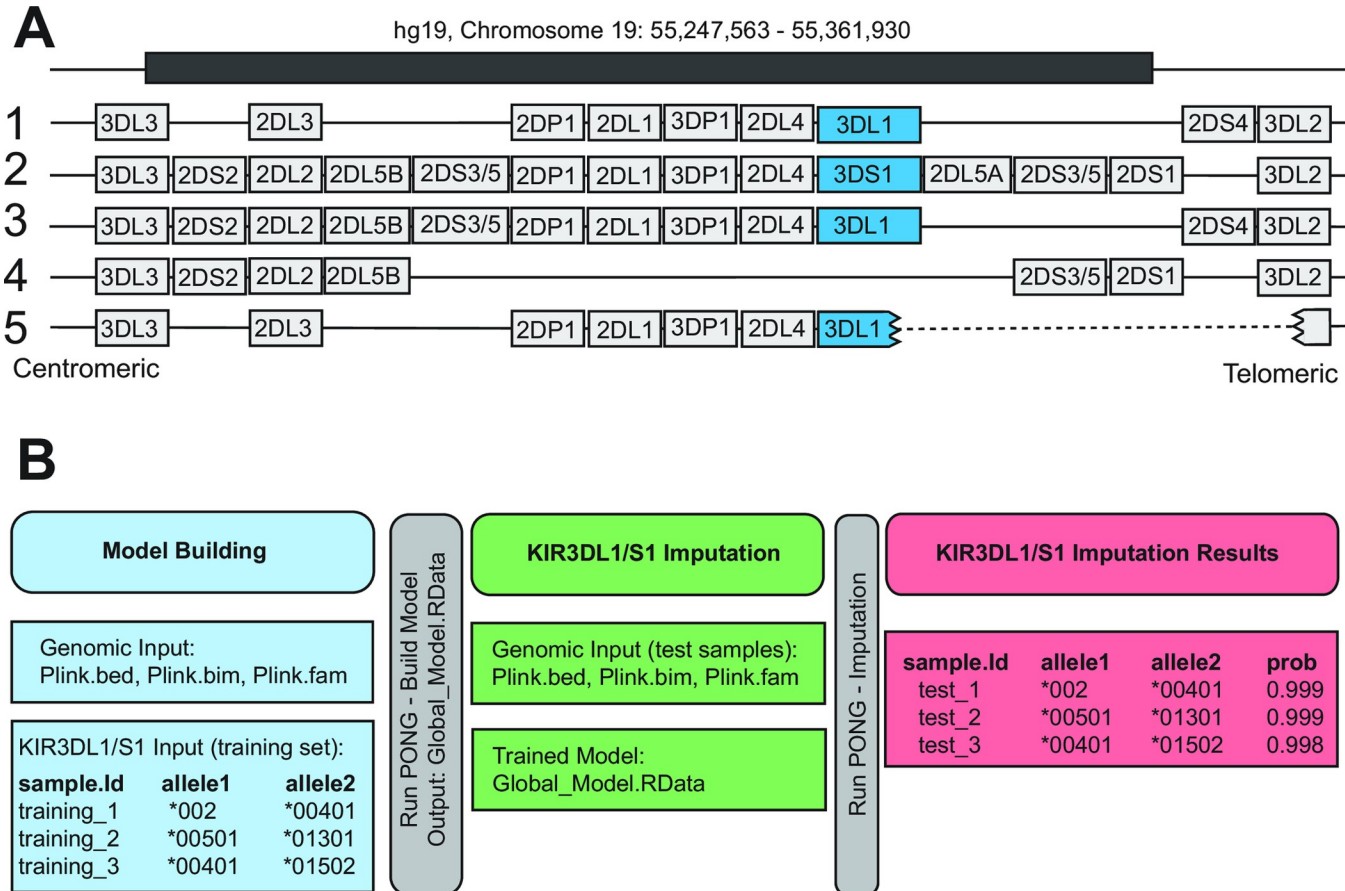

**Fig 1. Genomic location of *KIR3DL1/S1* and overview of allele imputation workflow. A.** Shows the location of the *KIR3DL1/S1* gene on five examples of *KIR* haplotypes. *KIR3DL1/S1* is shaded in blue, and other *KIR* genes are shaded grey. The *KIR3DL1/S1* gene can be absent (haplotype 4) or fused in-frame with *KIR3DL2* (haplotype 5) [92]. The human genome coordinates (build hg19) from which classifiers were drawn for imputation are given at the top. **B.** Schematic of model building, testing and output for the imputation of *KIR3DL1/S1* alleles using PONG. Shown are the required input files and their format for model building (blue) and testing (green). Red boxes give an example of the output from the imputation.

data [79] and because high-density SNP data (Illumina Omni 2.5 chip) is available from this cohort [78]. We distributed the 1,000 Genomes individuals into the designated five major population groups: Africa and African-descent (AFR), Americas (AMR), East Asia (EAS), Europe (EUR) and South Asia (SAS). We first optimized the model building parameters using the EUR group. We randomly divided each population group into two parts, building an imputation model using the first part and testing the imputation accuracy using the second part. We then built a global model by combining all of the 1,000 genomes individuals and repeating the process. We established that we could impute *KIR3DL1/S1* alleles using data from multiple low-density SNP genotyping platforms. Finally, we tested the efficacy of PONG using an independent population having both high-resolution sequence and WG-SNP data.

## Samples and genomic data

We obtained high density SNP data for chromosome 19, which contains the *KIR* genomic region (chromosome 19: 55247563–55361930, hg19), from the 1,000 Genomes Project Phase 3 individuals [78]. The data had been generated using the Illumina Omni 2.5 platform, which has 4,093 SNPs in the *KIR* genomic region [78]. To determine the *KIR3DL1/S1* alleles present

in each individual we used the Pushing Immunogenetics to the Next Generation (PING) pipe-line on high-depth sequence data, as previously described [79]. In total, there were 2,082 indi-viduals from the 1,000 Genomes data set from whom we had independently derived *KIR* sequence and chromosome 19 SNP data available (S1 Table), and these were divided into the designated five major population groups as indicated (S2 Table). We also analyzed SNP data obtained using the Infinium Immunoarray 24v2 (aka Immunochip) [80] from 397 Norwegians [81], from whom we determined high resolution *KIR3DL1/S1* genotypes (S3 Table) through targeted sequencing [79].

## Modifications to HIBAG to impute *KIR3DL1/S1*

We modified the HIBAG package version 1.2.4. The package name and relevant C++ functions were changed from HIBAG to KIRpong to avoid any conflict when both programs are installed. We removed genome build hg18 and included hg19 and GRCh38 (hg38) instead. We maintained many of the HIBAG functions while adjusting the selected chromosome posi-tions to target the *KIR* gene cluster on chromosome 19. We modified the 'hlaBED2Geno' func-tion to sample chromosome 19 positions 50247563–59128983 for hg19 and 46457117–58617616 for hg38. The 'hlaLociInfo' function was updated to target the *KIR* gene cluster and was specified as 55247563–55361930 for hg19 and 54734034–54853884 for hg38. Also in this function, the name of the gene was changed to *KIR3DL1/S1*. We also created PONG–Extended Window in which the 'hlaBED2Geno' function to sample chromosome 19 positions 55,100,000–55,500,000 (hg19) that can be used with low density SNP arrays to increase the size of the genomic region from which classifiers can be established. Finally, the printout messages were changed from HIBAG and *HLA* to PONG and *KIR3DL1/S1* to avoid confusion if both programs are active. The HIBAG functions were maintained, as extensive documentation for these functions is available.

## Optimization and testing of model building

The input data for model building is a text file containing the *KIR3DL1/S1* allele information, and SNP data in PLINK [82] binary format (.bed,.bim,.fam files) from the same individuals (Fig 1B). The first column of the text file contains the sample name (SampleID), the second column, *KIR3DL1/S1* allele 1 (Allele1) and the third column, allele 2 (Allele2). We optimized the model parameters using the 1,000 Genomes European populations group (EUR), compris-ing 353 individuals from five countries [78] and having 26 distinct *KIR3DL1/S1* alleles [79]. We chose the EUR subset to develop the filtering thresholds because this was the group having the fewest number of *KIR3DL1/S1* alleles before filtering, and because Europeans are currently the most extensively characterized of any major population groups [83–86]. We then expanded model building and testing to include populations from Africa (AFR, 558 individuals, 46 dis-tinct *KIR3DL1/S1* alleles), Hispanic/Latino populations from the Americas (AMR, 298 individ-uals, 34 *KIR3DL1/S1* alleles), East Asia (EAS, 406 individuals, 28 *KIR3DL1/S1* alleles) and South Asia (SAS, 467 individuals, 30 *KIR3DL1/S1* alleles).

We randomly selected 50% of individuals from the EUR group to be used for model build-ing. The remaining 50% of individuals were used to test the accuracy of the model. We first optimized the parameters to be used for filtering SNPs prior to model building. We compared the imputation accuracy of models built after removing SNPs with minor allele counts (MAC) $< 2$, or $< 3$, or a minor allele frequency (MAF) $< 1\%$ or $< 5\%$. We also tested the impact of removing individuals carrying any *KIR3DL1/S1* allele having MAC $< 3$ in the full EUR group (model + test). Once a robust model was established for the EUR population, we expanded the model to include all populations using the pre-filtering parameters and procedures established

above. Each model was evaluated based on the time needed for model building as well as the accuracy of imputation.

### Imputing *KIR3DL1/S1* alleles from low density SNP arrays

The next goal of this study was to test the accuracy of *KIR3DL1/S1* imputation using data from commonly used low density SNP arrays. From the 1,000 Genomes sequence data vcf files [78], we extracted those chromosome 19 SNPs equivalent to the Immunochip (4,902 SNPs, 26 in *KIR* region), Infinium (23,117 SNPs, 18 in *KIR* region) or MEGA (36,534 SNPs, 76 in *KIR* region) arrays. To increase the number of available classifiers, we built a second Global model in which we expanded the target region to chr19: 55,100,000–55,500,000 (hg19). These coordinates match those of the KIR*IMP program, which can be used to impute *KIR* gene content genotypes [87]. We then built and tested *KIR3DL1/S1* allele imputation models as described above. We used the Michigan imputation server [88] to supplement the low-density data with Illumina Omni 2.5 array data (1,832,506 chromosome 19 SNPs, 4,093 in *KIR* region). Input data was first phased with EAGLE [89] and an RSQ filter of 0.3 was applied to ensure imputation accuracy [90].

### Imputing *KIR3DL1/S1* alleles from an independent dataset

We also tested the accuracy of *KIR3DL1/S1* imputation using DNA sequence and WG-SNP data from 397 individuals from Norway. The cohort contained 18 distinct *KIR3DL1/S1* alleles (S3 Table), 14 of which were also present in the 1,000 Genomes data set. The cohort was genotyped using the Immunochip array, which has 26 SNPs in the *KIR* region, 16 of which are polymorphic in the data set. To increase model classifier density, we supplemented the SNP data with that of the Illumina Omni 2.5 array using the Michigan imputation server as described above. This process produced 715 SNPs in the *KIR* region that are variable in the data set. We also tested the extended genomic window (chr19: 55,100,000–55,500,000, hg19) for model building and this produced 3,440 variable SNPs in the Norway data set.

### Computational resources

All experiments were performed using a server with 512 GB 2400MHz RAM, running Ubuntu 18.04, R 3.5.1, R-server 1.1.456, and using a single core from a 2.3GHz Xeon E5-2697 CPU.

### Evaluation of imputation models

Overall accuracy of a given imputation model was determined as the number of correct allele calls made per individual (0, 1 or 2) divided by 2N. Sensitivity and specificity of a given model were determined per *KIR3DL1/S1* allele. Sensitivity was measured as the percentage of individuals known to be positive for a given allele who were also called positive for that allele by imputation. Specificity was determined as the percentage of individuals known to be negative for a given allele that were also called as negative for that allele by imputation.

## Results

### Parameters for frequency filtering of SNP and allele data

We designed, tested and optimized a model to impute *KIR3DL1/S1* alleles from WG-SNP data using a modification to the HIBAG framework and algorithm [75]. We used SNP data from the 1,000 Genomes project [78] and *KIR3DL1/S1* genotypes that we had previously determined from sequence data from the same individuals [79]. We focused first on the EUR group, comprised of 353 individuals and having 26 distinct *KIR3DL1/S1* allele sequences, ranging from

0.14% to 20% allele frequency (Fig 2A). We randomly selected 50% of the EUR individuals to be used for model building and used the other 50% to test the accuracy of the model. With the goal of maximizing the imputation accuracy of the test dataset, while preserving computational efficiency in model building, we first determined the effect of removing low-frequency SNPs. We measured the imputation accuracy of models that were built following removal of SNPs having a minor allele count (MAC) of < 2 (1,286 SNPs remaining in the *KIR* region) or MAC < 3 (1,044 SNPs remaining in the *KIR* region) in the full set of 353 individuals. We also measured the accuracy following removal of SNPs with a minor allele frequency (MAF) < 1% (941 SNPs remaining in the *KIR* region) or < 5% (645 SNPs remaining in the *KIR* region) in the full set of 353 individuals. A model was also built with no filtering of the genotype data for comparison (4,089 SNPs in *KIR* region). The *KIR* region SNPs used for testing the model accuracy were not filtered, and the models took from 6–9 seconds to impute *KIR3DL1/S1* alleles from the test data set of 177 individuals.

For the purposes of this test, accuracy was determined as the percentage of correct allele calls per individual in the test set. The lowest imputation accuracy, with 91% of individuals genotyped correctly, was obtained when no filtering was used or using a MAF < 5%. Models built using all other filtering parameters imputed the genotypes with 92% accuracy (Fig 2B and S4 Table). Thus, imputation accuracy was similar across all SNP frequency filtering parameters tested (Fig 2B). The model building run time ranged from 29 minutes, when SNPs were filtered at MAF < 5%, to 84 minutes when no filtering was used. We selected MAC < 3 as this was the fastest build time (66 minutes) for models of 92% accuracy (Fig 2B). Of the 26 *KIR3DL1/S1* alleles observed in the full EUR group (N = 353), twelve were observed less than three times (Fig 2A). Following removal of individuals possessing at least one of these twelve infrequent alleles, 14 *KIR3DL1/S1* alleles and 339 individuals remained in the population. As above, we removed SNPs having MAC < 3, divided this population in half, built a model and tested it on the other half. Compared with using MAC < 3 alone, the time required to build this model decreased from 66 minutes to 46 minutes, and the time to run the model reduced to 5 seconds (165 individuals), whereas the imputation accuracy increased from 92% to 96% (Fig 2B). Thus, this combination of filtering parameters produced the fastest time for model building and running, with the highest accuracy for imputing *KIR3DL1/S1* genotypes. Moreover, these parameters produced the greatest imputation accuracy across all five major population groups (S5 Table). We therefore implemented these parameters (MAC < 3 both for SNPs and *KIR3DL1/S1* alleles) in all subsequent analyses.

We next evaluated the sensitivity and specificity of the final EUR imputation model. Of 14 alleles in the model data set, 13 were also present in the test set (Fig 2C). We observed a mean specificity of 99%. The two alleles having < 100% specificity were *KIR3DL1*00101* and *KIR3DS1*01301*, with 98% specificity, thus for every 100 individuals imputed to have either allele, two were not shown as present through sequencing. We observed a mean sensitivity of 77%. All alleles with a frequency in the EUR group greater than 1.6% were imputed with >99% sensitivity. Below 1.6% allele frequency, two alleles (*3DL1*00402* and *3DS1*049N*) were imputed with 50% sensitivity and two (absence of *KIR3DL1/S1* (*00000) and *3DL1*009*) with 0% sensitivity. *KIR3DL1*00402* and *3DS1*049N* are each distinguished from their closest (parental) alleles by a single or a doublet nucleotide substitution, respectively [30]. Accordingly, in each case these alleles were imputed as the parental allele. *KIR3DL1*009* represents a double recombination having exons 2–3 identical to *3DS1*01301* and exons 1 and 4–9 identical to *3DL1*001* [91, 92]. The haplotype that lacks *KIR3DL1/S1* represents a large-scale deletion encompassing up to seven *KIR* genes (Fig 1A), and likely has very few identifying SNPs within the *KIR* locus. Thus, we observe a clear relationship between *KIR3DL1/S1* allele frequency and accuracy of imputation, with all high-frequency alleles being imputed with high

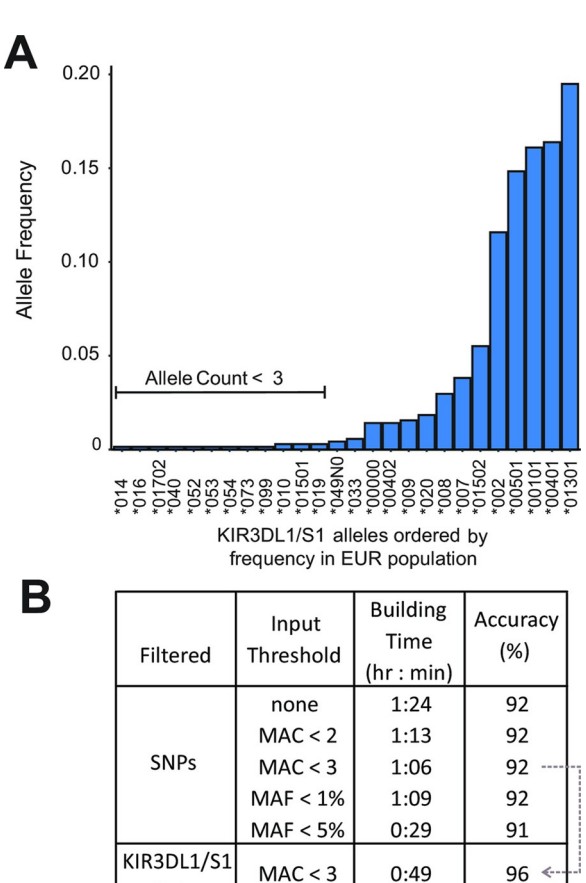

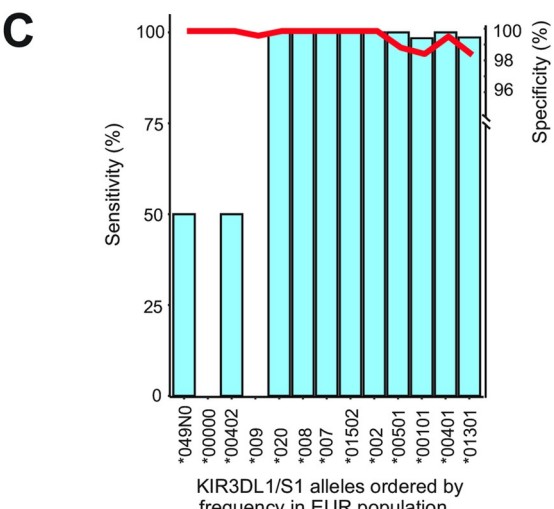

**Fig 2. Optimization of *KIR3DL1/S1* allele imputation using data from Europeans. A.** Bar graph shows the *KIR3DL1/S1* allele frequencies in the combined EUR population group [78] comprised of 353 individuals from Italy, Finland, United Kingdom, Spain, or Utah. The alleles were determined from short read sequence data [79]. **B.** Shown is a summary of the results obtained using models tested during optimization. From left to right are the filtered criteria (SNPs or *KIR3DL1/S1* alleles), the filtering threshold values, resulting model build time, and accuracy of the imputed genotypes. Grey dotted arrow indicates that the final model was built using MAC < 3 for SNPs and for *KIR3DL1/S1* alleles. **C.** Shows the imputation accuracy for each *KIR3DL1/S1* allele present in the final filtered EUR data set. Blue bars indicate the sensitivity (% of times a given allele was called as present when known to be present). Red line indicates specificity (% of times a given allele was called as absent when known to be absent).

accuracy, and those imputed with lower accuracy attributed both to their low frequency and lack of additional identifying characteristics.

## Development of a trained Global model for *KIR3DL1/S1* imputation

After establishing the most robust filtering parameters for model building in the EUR population group, we expanded the analysis to the four other major population groups from the 1,000 Genomes Project (Africa—AFR, Americas—AMR, East Asia—EAS and South Asia—SAS). We also combined all five population groups to form an additional 'Global' group (Table 1). As above, *KIR* locus SNPs and *KIR3DL1/S1* alleles having MAC < 3 in each respective group were removed, imputation models were then built using 50% individuals, and tested on the remaining 50%. Following the filtering based on *KIR3DL1/S1* allele counts, the African population group had the highest diversity with 31 alleles and the East Asian group the lowest with 13 alleles (Fig 3A). A total of 90 distinct *KIR3DL1/S1* alleles were present in the Global group, 42 of these occurred more than twice in total and were therefore included in model building. This allele filtering process resulted in 58 of the 2,082 individuals being removed. The Global model included 1,017 individuals and took 10 weeks to build. This process also increased the number of target alleles within all the individual population groups (Fig 3A).

In testing models built within each respective population group, imputation accuracy ranged from 80% in the AFR group to 96% in the EAS group (Fig 3B and S4 Table). When using the Global model, however, imputation accuracy increased for all groups, ranging from 88% in the AFR group to 97% in EAS (Fig 3B). When the test group was comprised of individuals drawn from all five of the population groups, an accuracy of 92% was achieved. This latter finding gives an estimate of the accuracy of *KIR3DL1/S1* allele imputation for individuals of unknown genetic ancestry. Using the Global model, the imputation time ranged from 2 min 9 s for AMR (N = 146) to 4 min 33 s for SAS (N = 228), and it took 16 min 1 s to impute the Global test set of 1,007 individuals (S4 Table).

We next evaluated the specificity and sensitivity of the Global imputation model. The mean specificity across the 42 alleles was 99%, with only two alleles having a specificity below 99% (Fig 3C). The lowest specificities were observed for *3DS1*01301* at 96% and *3DL1*01502* at 98.5%. Of the individuals falsely imputed as having *3DS1*01301*, 84% were due to a *KIR3DL1/S1* deletion haplotype. This finding is consistent with the suggestion that the parental haplotype for the deletion carried *3DS1*01301* [92]. The individuals falsely imputed as having *3DL1*01502*, possessed either *3DL1*01702*, *051* or *025* (33% each). All these alleles fall into the same ancestral lineage as *3DL1*015*, and likely exhibit similar phenotypes of high expression and ligand binding [1]. In the final Global population group (2N = 4,068) there were 15 *KIR3DL1/S1* alleles with a frequency above 1% and 27 alleles with a frequency below 1%

**Table 1. Number of *KIR3DL1/S1* alleles and individuals in data sets.**

| 1000 Genomes Population Group | Number of Individuals in Data Set | | | |
|---|---|---|---|---|
| | All | Global *KIR3DL1/S1* MAC < 3 | In Model Set | In Test Set |
| Africa (AFR) | 558 | 541 | 272 | 269 |
| Americas (AMR) | 298 | 292 | 146 | 146 |
| East Asia (EAS) | 406 | 389 | 196 | 193 |
| Europe (EUR) | 353 | 345 | 174 | 171 |
| South Asia (SAS) | 467 | 457 | 229 | 228 |
| Global | 2,082 | 2,024 | 1,017 | 1,007 |

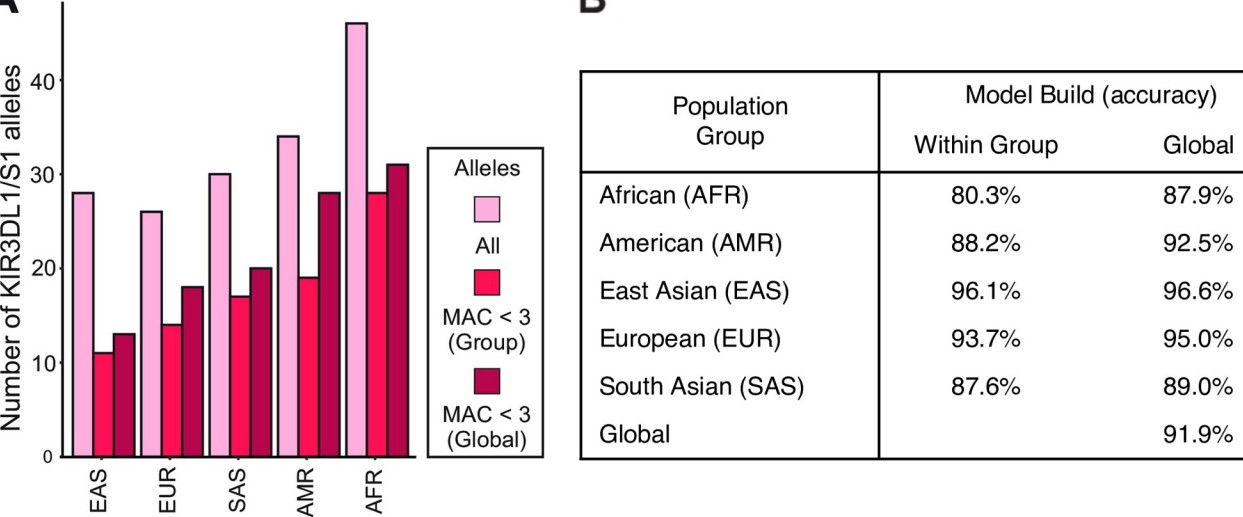

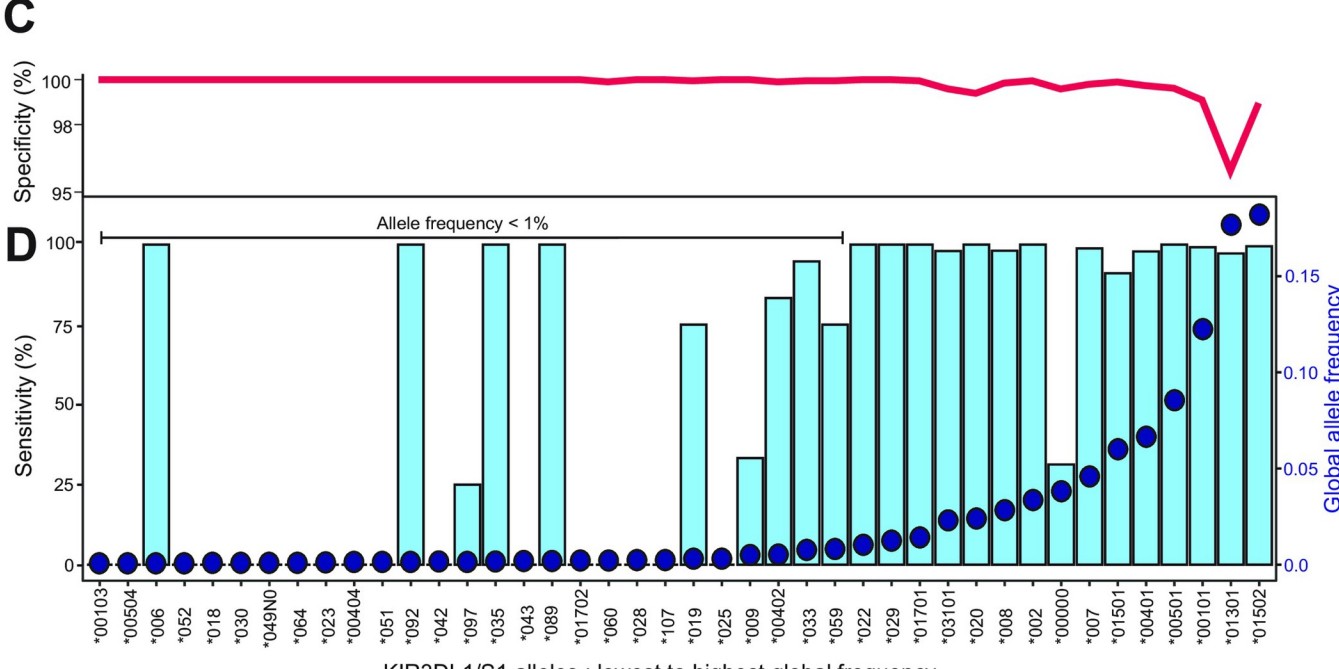

**Fig 3. Accurate imputation of *KIR3DL1/S1* alleles using a Global population model. A.** Bar graphs shows the number of *KIR3DL1/S1* alleles present in each of the five broad population groups of the 1,000 Genomes database. The bar colors indicate: (pink) the number of alleles present before filtering, (ruby) by MAC < 3 filtering, and (burgundy) by combining the five groups to form a Global population and then MAC < 3 filtering. The population groups are East Asian (EAS), European (EUR), South Asian (SAS), American (AMR) and African (AFR). **B.** Shows the imputation accuracy obtained for each of the population group and the Global models. (Within group) the model was built using 50% of the indicated group and tested on the other 50%. (Global) the model was built using 50% of all individuals and tested on the remaining 50% of the specified group. **C.** and **D.** Show the imputation efficacy for each allele present in the final Global data set. Blue bars indicate the sensitivity (% of times a given allele was called as present when known to be present). Red line indicates specificity (% of times a given allele was called as absent when known to be absent). Blue dots indicate the KIR3DL1/S1 allele frequencies in the Global population.

(Fig 3C). For those *KIR3DL1/S1* alleles having allele frequency below 1%, we observed a mean sensitivity of 29%. Despite a frequency of less than 1%, the alleles *006, *092, *035 and *089 were imputed with 100% sensitivity (Fig 3C). Conversely, *KIR3DL1/S1* alleles with a frequency

above 1% had a mean sensitivity of 95%. The mean sensitivity rose to 99% when the allele representing the absence of *KIR3DL1/S1* was excluded.

In total 27 *KIR3DL1/S1* alleles had a frequency less than 1% in the Global population. When the global frequency was above 1%, PONG was able to impute the alleles 91–100% of the time (Fig 3D). Thus, similar to the model built using the European population group, low frequency alleles were more likely to be incorrectly imputed than high frequency alleles. An exception to this was the allele representing the absence of *KIR3DL1/S1* (*00000) in which the frequency was 4% but PONG was only able to impute the absent allele correctly 35% of the time. Together these results show that PONG is effective for imputing common *KIR3DL1/S1* alleles across a diverse set of human populations, as well as some rare alleles. The AFR, AMR and SAS population groups had the highest number of *KIR3DL1/S1* alleles of frequency less than 1% in the test set (16, 10 and 10, respectively). Contributing to the lower accuracy of PONG in AFR and SAS, these low-frequency *KIR* alleles accounted for over 50% of those present in each group. By contrast, only three such low frequency alleles were present in the EAS group. In summary, the accuracy of PONG is affected by the frequency of *KIR3DL1/S1* alleles and is therefore less effective in more diverse human populations given a similar-sized training sample. Because we have filtered on minor allele count, the imputation accuracy in these populations will increase with the size of the model data set used.

## Imputing *KIR3DL1/S1* alleles using low density genotyping datasets

We evaluated whether *KIR3DL1/S1* genotyping could be performed using data obtained from each of three commonly used low density arrays. For this test, we included the AFR and EAS groups because they represent the highest and lowest genetic diversity, respectively, of the five major population groups in the 1,000 Genomes database. We based this test on the Global model and extended the window from which classifiers are sampled to chr19: 55,100,000–55,500,000 (hg19). To ensure continuity, we used the same individuals as the Global model development described above. Using the SNP genotype data directly produced poor results for each of these chips (Table 2). Thus, we first supplemented the data by imputing Illumina Omni 2.5 array SNPs. Accuracy in the AFR population improved to 78%, 80% and 84% when the genotypes originated from Infinium, Immunochip or MEGA SNPs, respectively (Table 2). For the EAS group the accuracy was 81%, 89% and 92% for Infinium, Immunochip and MEGA arrays, respectively (Table 2). As expected, the accuracy was lower than that obtained directly from the high-density Illumina Omni 2.5 data (88% AFR and 97% EAS, Fig 3). That we observed little difference for the EUR group between low- and high-density arrays, with a minimum of 91% accuracy, may be due to original ascertainment bias in chip design.

**Table 2. Accuracy of PONG across SNP arrays.**

| Test Set Parameters | | Accuracy (% of Genotypes Imputed Correctly) | | | |
|---|---|---|---|---|---|
| Population | SNP Imputation[+] | Illumina Omni 2.5 | Infinium | Immunochip | MEGA |
| EUR | no | 94% | 49% | 55% | 58% |
| | yes | – | 91% | 91% | 92% |
| EAS | no | 96% | 61% | 72% | 78% |
| | yes | – | 81% | 89% | 92% |
| AFR | no | 88% | 28% | 54% | 37% |
| | yes | – | 78% | 80% | 84% |

Genome window chr19: 55,100,000–55,500,000 (hg19).

[+] Model and test sets supplemented with imputed Illumina Omni 2.5 genotypes.

Consistent with this reasoning, the MEGA chip, which was designed to reduce such bias [93], gave the greatest accuracy through this test. In summary, these results show it is tractable to use PONG with low-density arrays when higher-density array genotypes are first imputed from the starting data.

## Testing PONG using an independent data set

We analyzed a cohort of 397 individuals from Norway, from whom we generated both Immunochip SNP and high-resolution *KIR3DL1/S1* allele sequence data. We observed a strong correlation between allele frequencies in Norway and the 1,000 Genomes EUR group (r = 0.96). In total, 18 *KIR3DL1/S1* alleles were identified in the Norwegian cohort through nucleotide sequencing. After filtering for MAC < 3, there were 13 *KIR3DL1/S1* alleles present. We built a model from this cohort and used it for the imputation of *KIR3DL1/S1* alleles. Imputation increased the number of variable SNPs from 26 to 715 in the *KIR* region, improving accuracy of *KIR3DL1/S1* allele imputation from 53% to 89%. We extended the window for model building to chr19: 55,100,000–55,500,000 (hg19), which contains 255 SNPs on the Immunochip. With the extended window we observed 92% accuracy, sensitivity of mode 100% and mean 75%, and specificity of mode 100%, mean 99% (Fig 4). As in previous analyses, *KIR3DL1/S1*

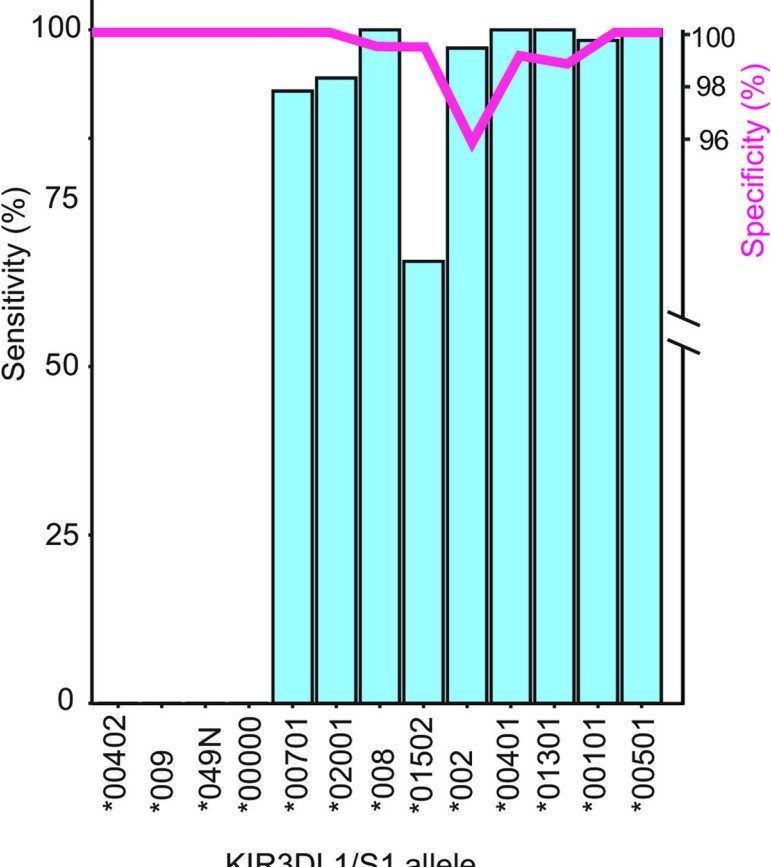

**Fig 4. Accurate imputation of *KIR3DL1/S1* alleles from Immunochip SNP data.** Bar graph shows the efficiency of *KIR3DL1/S1* allele imputation using a model built and tested on a cohort from Norway who also had their *KIR3DL1/S1* alleles genotyped to high resolution. Blue bars indicate the sensitivity (% of times a given allele was called as present when known to be present). Red line indicates specificity (% of times a given allele was called as absent when known to be absent).

alleles having frequency > 1% have greater imputation accuracy than those < 1% (Fig 4). In this analysis, the mean sensitivity of alleles with a frequency of < 1% was 33%. By contrast, *KIR3DL1/S1* alleles with a frequency > 1% had a mean sensitivity of 85%. We note that, consistent with the 1,000 Genome EUR test above, imputation of SNPs in the extended window prior to model building did not increase the accuracy in this cohort. These experiments further demonstrate that high resolution *KIR3DL1/S1* genotypes can be imputed from low-density SNP arrays, and with similar accuracy to high-density arrays.

## Running PONG

The PONG program is installed using the command line and opened as a library in R (R version 2.14.0–4.0.0.) [94]. PONG can be run using WG-SNP data mapped either to hg19 or hg38. The HIBAG imputation algorithm does not require data to be phased [75]. The Global model (hg19) and the model built with the EUR group (hg38) are available for download. Other models will be added as they become available. Using our Global model, we estimate that 1,000 individuals could be imputed every 15 minutes using a single core on a laboratory server, such as the one we have used. Users can also create their own models when WG-SNP data and *KIR3DL1/S1* allele genotypes are available, and modify the data input and filtering parameters, as described in the tutorial.

## Discussion

Knowledge of *KIR3DL1/S1* diversity can help predict the course of specific infections, immune-mediated diseases, and their therapies [95–101]. However, by virtue of the polymorphic and structural complexity at the locus, it is often excluded from genome-wide association studies. The primary goal of this study was to develop a model trained to impute *KIR3DL1/S1* alleles rapidly from WG-SNP data encompassing a wide range of human genetic diversity. We built imputation models using high-density WG-SNP data [78] and high-resolution *KIR3DL1/S1* allele calls [79] from the five broadly defined 1,000 Genomes population groups, and then built a model for the Global group. To achieve these goals, we adapted the coding framework and algorithm from HIBAG [75] in a modification that we have named PONG. We determined that the imputation models are most effective when both the WG-SNP data and *KIR3DL1/S1* alleles have been filtered to remove alleles that occur infrequently. The former filter to reduce the model building run time and the latter to increase imputation accuracy. The resulting range of imputation accuracies of the final Global model was 88% for Africans and South Asians, to 97% for East Asians. The 1,000 Genomes WG-SNP data has a dense set of genotypes, including 1,832,506 SNPs from chromosome 19 [78]. Other genotype chips used for disease association studies have less dense sets of SNPs, including the Infinium Immunoarray, which targets markers associated with autoimmune disease and inflammatory disorders [80]. Because *KIR3DL1/S1* diversity is associated with development or severity of multiple autoimmune diseases [20, 22, 23, 25, 102], we tested the accuracy of imputation using results generated from this genotyping chip and achieved similar imputation accuracy as achieved from the high-density array.

Although PONG is effective in imputing *KIR3DL1/S1* alleles, there are a few limitations to this program that we are optimistic will improve over time. As observed for *HLA* [103], we found a negative correlation between the accuracy of PONG and the diversity of *KIR3DL1/S1* alleles in a population. For example, the African population group we studied has the highest number of distinct *KIR3DL1/S1* alleles as well as the highest number with an allele frequency below 1%. We also observed a high incidence of alleles at less than 1% in the South Asian group, as well as a higher frequency of haplotypes having duplicated *KIR3DL1/S1* compared

with Africans [92]. The result is that imputation accuracy is lowest in Africans and South Asians. Conversely, the East Asian group has the lowest number of *KIR3DL1/S1* alleles of allele frequency below 1%, and the highest imputation accuracy. Therefore, PONG is most effective at imputing the most frequent alleles. Finally, *KIR3DL1/S1* allelic diversity is under-characterized for many non-European origin populations. Any *KIR3DL1/S1* alleles newly identified in these populations will need to be incorporated into future models to increase imputation accuracy. The imputation accuracy of the model will improve over time as more immunogenetic studies of *KIR3DL1/S1* are conducted, thus expanding our sample set for building more robust and diverse models. Given that the model is open source, and that PONG has a model building function available, this can be achieved both by the developers and users.

PONG is less accurate imputing the absence of the *KIR3DL1/S1* gene (which we designated *00000), and as we were only able to impute this null *KIR3DL1/S1* allele at an accuracy of 35% using the Global model. One solution to improve the accuracy of imputing the '*KIR3DL1/S1* absent' allele may be to couple the use of PONG with the program KIR*IMP. KIR*IMP is targeted to *KIR* gene content diversity and can impute the presence or absence of *KIR3DL1/S1* with an accuracy above 90% [87]. KIR*IMP does not require installation and is run through a web interface, via data upload. Accurate sequencing and assembly can be challenging for highly polymorphic and structurally diverse regions of the genome [104]. Both these phenomena are characteristics of the *KIR* locus [29]. Therefore, PONG relies on high quality WG-SNP data with robust quality control measures implemented in SNP calling pipelines. New techniques to improve the identification of structural variation are being created, including long-range optical mapping, which uses the optical signal strength from each SNP genotype to identify deletions and duplications [105]. Together, an increased sampling of individuals having rare *KIR3DL1/S1* alleles and better characterization of structural variation from WG-SNP data will likely improve the imputation accuracy of PONG. PONG is open source and gives users the ability to publish reference models that do not contain information about individuals, nor is the upload of genotype data to any public server required. These features both support reproducibility of research and increase efficiency of the imputation as the models are expanded.

Highly polymorphic interactions of KIR3DL1/S1 with HLA-A and -B modulate the critical functions of NK cells in immunity, which include the destruction of infected or cancerous cells [2]. Combinatorial diversity of KIR3DL1/S1 with HLA-A and -B allotypes thus affects the susceptibility and course of multiple immune-mediated diseases. Several methods are available to impute *HLA* alleles [72–75], but large-scale genetic studies often exclude analysis of *KIR3DL1/S1* due to the exceptional polymorphism and structural diversity of the genomic region. A secondary goal of this study was thus to produce imputation models that could be used in conjunction with existing models to impute the combinatorial diversity of KIR3DL1/S1 and HLA allotypes. By comparison with *KIR3DL1/S1*, the mean imputation accuracy for HIBAG across seven *HLA* genes was 81.2% in African populations and 91.1% in East Asians [75]. In African populations, *HLA-DPB1* had the lowest imputation accuracy at 74.2% and *HLA-A* had the highest observed accuracy at 92.4%. The corresponding imputation accuracies of these *HLA* genes in East Asians were 89.8% and 92.1% respectively. Whereas HIBAG did not originally include South Asians, subsequent studies in a population from India found an imputation accuracy of 88% for *HLA-B* rising to 94% for other *HLA* genes [90]. The mean accuracy of *KIR3DL1/S1* allele imputation described herein is equivalent, and often better than that obtained for *HLA class I* and *II* using the same underlying algorithm. We therefore propose that using this algorithm to impute both *KIR3DL1/S1* and *HLA-A* and *-B* genotypes from WG-SNP data presents a considerable advantage over other approaches. This approach is particularly applicable for studies of Biobank data, where targeted sequencing of *KIR3DL1/S1* and

*HLA-A* and *-B* from many thousands of individuals is not currently tractable. Utilizing our pre-built models, PONG can be implemented to make genetic association studies of *KIR3DL1/S1* in combination with *HLA-A* and *-B* possible at very large scale.

Due to their established associations with multiple human ailments including infectious disease, cancer and autoimmunity [1], we focused this study on determining the allotypes of *KIR3DL1/S1*. Included among those with the greatest imputation accuracy are common allotypes having altered or no expression, specificity-determining polymorphism, and an activating variant [2]. Any of these allotypes can have dramatic consequence for the function of NK cells. Thus, knowledge of *KIR3DL1/S1* genotype, in combination with *HLA-A* and *-B* genotypes, is often sufficient to understand the disease mechanism or move the field forward in other ways [15, 51, 102, 106]. In other cases, it will be necessary to form a complete picture of KIR interactions with HLA-C in addition to -A and -B, by generating a genotype from the entire *KIR* locus [29]. Validation of PONG to impute *KIR3DL1/S1* is a robust proof of concept that can be built upon to impute entire *KIR* region genotypes. Expanding to other *KIR* genes will be challenging as the genomic region contains substantial structural variation and, unlike *HLA*, classifiers are sampled from the entire locus and flanking parts, rather than individual genes. We conducted rigorous testing to determine the efficacy and limits of HIBAG in imputing *KIR3DL1/S1* genotypes. This was a first but important step before expanding the concept in future work to impute complete *KIR* haplotypes.

## Supporting information

**S1 Table. *KIR3DL1/S1* Genotypes of 1,000 Genomes Individuals.** This table contains the *KIR3DL1/S1* alleles corresponding to each of the individuals present in the 1000 Genomes database that were used in this study. The population designation and sample ID is present for each.
(XLSX)

**S2 Table. Summary of the 1000 Genomes populations utilized in this study.** This table shows the number of individuals from each sub-population used to develop PONG, arranged by population group.
(XLSX)

**S3 Table. *KIR3DL1/S1* Genotypes of Norwegian Individuals.** This table shows the *KIR3DL1/S1* genotypes used for testing the independent data set from Norway.
(XLSX)

**S4 Table. Parameters and Output statistics of Imputation Models.** This table provides a summary of the results obtained through each experiment conducted to develop the optimal parameters for PONG model building, as well as testing the accuracy. The filtering parameters, populations used for both model building and testing, genotype array, accuracy, and time for model building and imputation are all available in this table.
(XLSX)

**S5 Table. Filtering parameter optimization for Imputation Models.** This table shows the percent accuracy of PONG for each of the main population groups as a function of distinct SNP and *KIR3DL1/S1* filtering parameters.
(XLSX)

## Author Contributions

**Conceptualization:** Damjan Vukcevic, Stephen Leslie, Paul J. Norman.

**Data curation:** Genelle F. Harrison, Laura Ann Leaton, Katherine M. Kichula, Marte K. Viken, Paul J. Norman.

**Formal analysis:** Genelle F. Harrison, Laura Ann Leaton.

**Funding acquisition:** Paul J. Norman.

**Investigation:** Genelle F. Harrison, Laura Ann Leaton, Paul J. Norman.

**Methodology:** Genelle F. Harrison, Laura Ann Leaton, Jonathan Shortt, Christopher R. Gignoux, Paul J. Norman.

**Project administration:** Paul J. Norman.

**Resources:** Marte K. Viken, Christopher R. Gignoux, Benedicte A. Lie, Paul J. Norman.

**Software:** Genelle F. Harrison, Laura Ann Leaton, Erica A. Harrison, Jonathan Shortt.

**Supervision:** Paul J. Norman.

**Validation:** Genelle F. Harrison, Laura Ann Leaton, Erica A. Harrison.

**Visualization:** Genelle F. Harrison, Paul J. Norman.

**Writing – original draft:** Genelle F. Harrison, Laura Ann Leaton, Paul J. Norman.

**Writing – review & editing:** Genelle F. Harrison, Laura Ann Leaton, Erica A. Harrison, Katherine M. Kichula, Marte K. Viken, Jonathan Shortt, Christopher R. Gignoux, Benedicte A. Lie, Damjan Vukcevic, Stephen Leslie, Paul J. Norman.

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
