## [Decision Letter · Decision Letter 0]

3 Aug 2021

Dear Dr. Harrison,

Thank you very much for submitting your manuscript "Allele imputation for the Killer cell Immunoglobulin-like Receptor KIR3DL1/S1" for consideration at PLOS Computational Biology.

As with all papers reviewed by the journal, your manuscript was reviewed by members of the editorial board and by several independent reviewers. In light of the reviews (below this email), we would like to invite the resubmission of a significantly-revised version that takes into account the reviewers' comments.

Note, in particular, that both reviewers suggest extending the imputation to include also other KIR genes or at least to justify why this is not done -- I think that the former option would be strongly preferable. 

We cannot make any decision about publication until we have seen the revised manuscript and your response to the reviewers' comments. Your revised manuscript is also likely to be sent to reviewers for further evaluation.

Sincerely,

Roger Dimitri Kouyos

Associate Editor

PLOS Computational Biology

Thomas Leitner

Deputy Editor

PLOS Computational Biology

Reviewer's Responses to Questions

**Comments to the Authors:**

Reviewer #1: This manuscript by Harrison and colleagues describes an interesting adaptation of the widely used HIBAG method for HLA allelic inference from SNP array data, making it possible to impute allelic variation of the KIR3DL1/DS1 gene. Given the demonstrated functional differences of KIR alleles with regard to expression and affinity to their HLA ligands, as well as allele-specific disease associations, the work is an important contribution and very useful for the immunogenetics research community. In contrast to available WGS- or WES-based KIR inference methods that rely on the availability of sequence-level information, PONG allows large-scale imputation of KIR3DL1/DS1 allelic data from cohorts with available array-based SNP genotyping data.

The reported KIR typing accuracy is high, and is comparable to commonly used HLA imputation methods. As the authors discuss, accuracy is slightly lower for populations with higher allelic diversity, which is indeed not unexpected and helps to make the case for increased research efforts in these populations.

The manuscript is well written, and the rationale clearly described.

Adapting the HIBAG method allowed the authors to provide the imputation models to the scientific community without having to share individual-level genetic data, which is often not possible due to IRB regulations and ethical concerns.

Major points:

KIR3DL1/DS1 is not the only KIR gene with functionally relevant allelic diversity.

Once the modified HIBAG method is established, it should be conceptually straightforward to also impute allelic diversity for other KIR genes. For example, KIR2DL3 and KIR2DL2 can have different binding affinities to their HLA-C C1 and C2 ligands depending on allele status. When reading the paper, I expected a discussion why other KIR genes were considered out of scope. Do you consider them less relevant? Are there significant issues related to the complexity of the region (copy number, etc.) that make this more difficult, and that I am not aware of (not unlikely, in which case I'd like to learn)? Are you going to tackle those next (yes, please)?

In its current form, it felt a bit like reviewing an HLA imputation method that only imputes e.g. HLA-B. Even if very accurate, it's important to consider downstream association results with disease phenotypes in the context of the haplotype structure. Since e.g. KIR3DL1 and KIR2DL3 are both present on the most common KIR A haplotypes, but in different combinations on KIR B haplotypes, inferring causality relies on the ability to correct for the presence / absence of other genes and alleles.

To be clear, I think PONG is definitely useful in its current form, and I will for sure use it myself, but I'm missing a good explanation for the narrow scope, given how much an addition of other genes would add in terms of usefulness.

Minor points:

- In the introduction, the authors mention HIV research and cancer therapy as key areas impacted by KIR/HLA diversity, but the cited references 43-46 only refer to HIV. Please add some cancer-specific references.

- It would be great to further assess differences in imputation accuracy from other SNP arrays with low marker density in the LRC region.

- The authors mention the possibility of using KIR*IMP in addition to improve gene presence / absence inference. I applaud the author's decision to make PONG freely available to the scientific community. Unfortunately, KIR*IMP doesn't offer the same license model. Also, individual-level genotype data needs to be uploaded to a server, which is often not possible due to IRB restrictions. I think the latter should be mentioned when suggesting the joint use of both methods.

- The authors might consider putting more emphasis on the advantages of the HIBAG method, allowing researchers to publish reference models for imputation without the requirement to share whole panels with individual-level genetic information, which can limit the usefulness of alternative methodological approaches.

Reviewer #2: The study by Harrison et al utilises a machine-learning approach to accurately imput alleles of the highly variable KIR3DL1/S1 gene from high density SNP data. This genetic system encodes proteins on the surface of NK cells and in combination with specific ligands (HLA; another polymorphic system) have been shown to be associated with a number of immune-mediated diseases. Given the complexity of the KIR genetic region on chr 19, high-resolution KIR genotyping is typically needed to evaluate this genetic system in disease association studies. This study describes a valuable computational tool to imput KIR3DL1/S1 genotypes based on genome-wide SNP data that is often available in genetic-based association studies. Furthermore, the accuracy of the tool (PONG) is comparable to levels obtained from other programs to imput HLA.

There are certain aspects that the authors should clarify regarding the analysis as indicated below:

i) In the methods section on page 8, it should be clear that the KIR genotyping determined by PING is based on sequence data. It is unclear from the text that this is the case and requires examination of reference 77. Furthermore, it is unclear what is the relevance of the 143 subjects with Sanger-based sequencing. If this was also used for KIR genotyping then this raises issues related to this approach including phasing.

ii) The use of a second SNP platform with a set of Norwegian subjects is useful to determine the value of PONG with a SNP array with a different density across the relevant region. However, could the the outcome may be an over-estimate given the likely higher linkage disequilibrium and reduced genetic diversity in this population relative to others such as an African cohort?

iii) There appears to be a discrepancy in the sensitivity values in the abstract and author's summary.

iv) The lower accuracy in the African subjects (89%) is as expected given the lower LD and diversity but can the authors explain the lower % for the South Asian subjects? Is this different to what is observed for the HLA imputation tools?

v) Can the authors justify the use of the EUR subjects to determine the threshold/filtering parameters?

vi) 3DL1/S1 is an important gene in the KIR complex but there are a number of other KIR genes in the region that also bind Bw4 and the C1/C2 ligands and are variable. Have the authors attempted to imput the genotypes of these other genes?

**Have the authors made all data and (if applicable) computational code underlying the findings in their manuscript fully available?**

Reviewer #1: Yes

Reviewer #2: Yes

PLOS authors have the option to publish the peer review history of their article (what does this mean?). If published, this will include your full peer review and any attached files.

Reviewer #1: **Yes: **Christian Hammer

Reviewer #2: No
---

## [Decision Letter · Decision Letter 1]

10 Jan 2022

Dear Dr. Harrison,

We are pleased to inform you that your manuscript 'Allele imputation for the Killer cell Immunoglobulin-like Receptor KIR3DL1/S1' has been provisionally accepted for publication in PLOS Computational Biology.

Best regards,

Roger Dimitri Kouyos

Associate Editor

PLOS Computational Biology

Thomas Leitner

Deputy Editor

PLOS Computational Biology

Reviewer's Responses to Questions

**Comments to the Authors:**

Reviewer #1: The authors have responded to all questions and comments in great detail, and I appreciate the additional context and background information, as well as the significant effort to add results for different arrays. I have no further questions or comments.

Reviewer #2: Authors have addressed the issues raised by the reviewers.

**Have the authors made all data and (if applicable) computational code underlying the findings in their manuscript fully available?**

Reviewer #1: Yes

Reviewer #2: None

PLOS authors have the option to publish the peer review history of their article (what does this mean?). If published, this will include your full peer review and any attached files.

Reviewer #1: **Yes: **Christian Hammer

Reviewer #2: No

---

## [Editor Report · Acceptance letter]

16 Feb 2022

PCOMPBIOL-D-21-00820R1 

Allele imputation for the Killer cell Immunoglobulin-like Receptor KIR3DL1/S1

Dear Dr Harrison,

I am pleased to inform you that your manuscript has been formally accepted for publication in PLOS Computational Biology. Your manuscript is now with our production department and you will be notified of the publication date in due course.

With kind regards,

Anita Estes
